# Bias Amplification Enhances Minority Group Performance

**Gaotang Li**[*]                                                                               *gaotang@umich.edu*
*University of Michigan*
*Ann Arbor, MI*

**Jiarui Liu**[*]                                                                               *jiaruil5@cs.cmu.edu*
*Carnegie Mellon University*
*Pittsburgh, PA*

**Wei Hu**                                                                                      *vvh@umich.edu*
*University of Michigan*
*Ann Arbor, MI*

**Reviewed on OpenReview:** *https://openreview.net/forum?id=75OwvzZZBT*

## Abstract

Neural networks produced by standard training are known to suffer from poor accuracy on rare subgroups despite achieving high accuracy on average, due to the correlations between certain spurious features and labels. Previous approaches based on worst-group loss minimization (*e.g.* Group-DRO) are effective in improving worse-group accuracy but require expensive group annotations for all the training samples. In this paper, we focus on the more challenging and realistic setting where group annotations are only available on a small validation set or are not available at all. We propose BAM, a novel two-stage training algorithm: in the first stage, the model is trained using a *bias amplification* scheme via introducing a learnable *auxiliary variable* for each training sample; in the second stage, we upweight the samples that the bias-amplified model misclassifies, and then continue training the same model on the reweighted dataset. Empirically, BAM achieves competitive performance compared with existing methods evaluated on spurious correlation benchmarks in computer vision and natural language processing. Moreover, we find a simple stopping criterion based on *minimum class accuracy difference* that can remove the need for group annotations, with little or no loss in worst-group accuracy. We perform extensive analyses and ablations to verify the effectiveness and robustness of our algorithm in varying class and group imbalance ratios.[1]

## 1 Introduction

The presence of spurious correlations in the data distribution, also referred to as "shortcuts" (Geirhos et al., 2020), is known to cause machine learning models to generate unintended decision rules that rely on spurious features. For example, image classifiers can largely use background instead of the intended combination of object features to make predictions (Beery et al., 2018).

Similar phenomenon is also prevalent in natural language processing (Gururangan et al., 2018) and reinforcement learning (Lehman et al., 2020). In this paper, we focus on the *group robustness* formulation of such problems (Sagawa et al., 2019), where we assume the existence of *spurious attributes* in the training data and define *groups* to be the combination of class labels and spurious attributes. The objective is to achieve high *worst-group accuracy* on test data, which would indicate that the model is not exploiting the spurious attributes.

---

[*]Equal contribution.
[1]Our code is available at `https://github.com/motivationss/BAM`

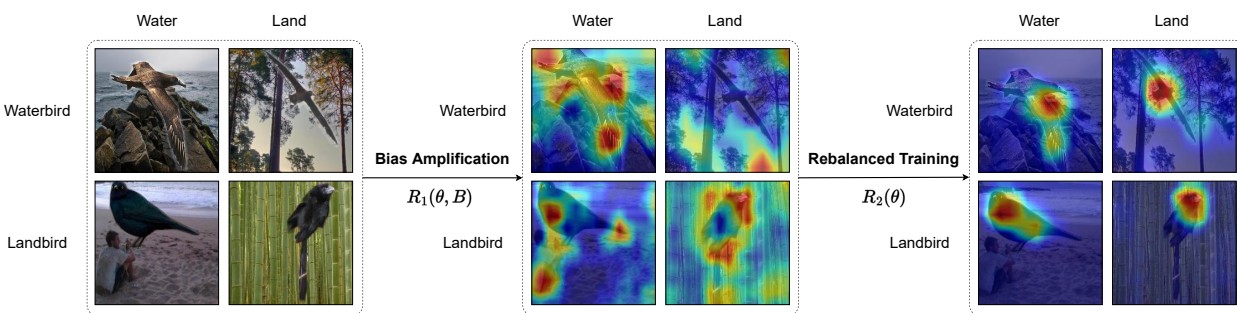

Figure 1: Using Grad-CAM (Selvaraju et al., 2017) to visualize the effect of bias amplification and rebalanced training stages, where the classifier heavily relies on the background information to make predictions after bias amplification but focuses on the useful feature (bird) itself after the rebalanced training stage.

Under this setup, one type of methods use a distributionally robust optimization framework to directly minimize the worst-group training loss (Sagawa et al., 2019). While these methods are effective in improving worst-group accuracy, they require knowing the group annotations for all training examples, which is expensive and oftentimes unrealistic. In order to resolve this issue, a line of recent work focused on designing methods that do not require group annotations for the training data, but need them for a small set of validation data (Liu et al., 2021; Nam et al., 2020; 2022; Zhang et al., 2022). A common feature shared by these methods is that they all consist of training two models: the first model is trained using plain empirical risk minimization (ERM) and is intended to be "biased" toward certain groups; then, certain results from the first model are utilized to train a debiased second model to achieve better worst-group performance. For instance, a representative method is JTT (Liu et al., 2021), which, after training the first model using ERM for a few epochs, trains the second model while upweighting the training examples incorrectly classified by the first model.

The core question that motivates this paper is: *Since the first model is intended to be biased, can we amplify its bias in order to improve the final group robustness?* Intuitively, a bias-amplified first model can provide better information to guide the second model to be debiased, which can potentially lead to improving group robustness. To this end, we propose a novel two-stage algorithm, BAM (Bias AMplification), for improving worst-group accuracy without any group annotations for training data:

- *Stage 1: Bias amplification.* We train a bias-amplified model by introducing a trainable auxiliary variable for each training example.

- *Stage 2: Rebalanced training.* We upweight the training examples that are misclassified in Stage 1, and continue training the same model instead of retraining a new model.[2]

Evaluated on various standard benchmark datasets for spurious correlations, including Waterbirds (Wah et al., 2011; Sagawa et al., 2019), CelebA (Liu et al., 2015; Sagawa et al., 2019), MultiNLI (Williams et al., 2018; Sagawa et al., 2019), and CivilComments-WILDS (Borkan et al., 2019; Koh et al., 2021), we find that BAM achieves competitive worst-group accuracy compared to existing methods in the setting where group annotations are only available on a validation set.

Digging deeper into the mechanism of BAM, we observe that auxiliary variables learned in Stage 1 exhibit clearly different magnitudes between majority and minority group examples, thus confirming their effectiveness in bias amplification. We also find that BAM achieves robust performance across hyperparameter choices. In addition, our ablation studies demonstrate the clear advantage of continued training in Stage 2 over training a separate model from scratch and the effectiveness of each of our proposed components.

Furthermore, we explore the possibility of *completely* removing the need for group annotations. We find that low class accuracy difference (which does not require any group annotations to evaluate) is strongly correlated

---

[2]In Figure 1, we use Grad-CAM visualization to illustrate that our bias-amplified model from Stage 1 focuses more on the image background while the final model after Stage 2 focuses on the object target.

with high worst-group accuracy. Using minimum class accuracy difference as the stopping criterion, BAM achieves the best overall performance on benchmark datasets in the complete absence of group annotations. We also perform extensive controlled experiments and find that this approach is robust across different datasets, varying dataset sizes, and varying imbalance ratios.

## 2 Related Works

A variety of recent work discussed different realms of robustness, for instance, class imbalance (He & Garcia, 2009; Huang et al., 2016; Khan et al., 2017; Johnson & Khoshgoftaar, 2019; Thabtah et al., 2020), and robustness in distribution shift, where the target data distribution is different from the source data distribution (Clark et al., 2019; Zhang et al., 2020; Marklund et al., 2020; Lee et al., 2022; Yao et al., 2022). In this paper, we mainly focus on improving group robustness. Categorized by the amount of information available for training and validation, we discuss three directions below.

**Improving Group Robustness with Training Group Annotations.** Multiple works used training group annotations to improve worst-group accuracy (Byrd & Lipton, 2019; Khani et al., 2019; Goel et al., 2020; Cao et al., 2020; Sagawa et al., 2020). Other works include minimizing the worst-group training loss using distributionally robust optimization (Group-DRO) (Sagawa et al., 2019), simple training data balancing (SUBG) (Idrissi et al., 2022), and retraining the last layer of the model on the group-balanced dataset (DFR) (Kirichenko et al., 2022). These methods achieve state-of-the-art performance on all benchmark datasets. However, the acquisition of spurious attributes of the entire training set is expensive and often unrealistic.

**Improving Group Robustness with Validation Group Annotations Only.** Acknowledging the cost of obtaining group annotations, many recent works focus on the setting where training group annotations are not available (Duchi & Namkoong, 2019; Oren et al., 2019; Levy et al., 2020; Pezeshki et al., 2021). Taghanaki et al. (2021) proposed a transformation network to remove the spurious correlated features from image datasets and then choose classifier architectures according to the downstream task. Shu et al. (2019) utilized a small set of unbiased meta-data to reweight data samples. CVaR DRO (Duchi et al., 2019) introduced a variant of distributionally robust optimization that dynamically reweights data samples that have the highest losses. Jeon et al. (2022) proposed a method that utilizes the multi-variant and unbiased characteristics of CNNs through hierarchical feature extraction and orthogonal regularization. In particular, the most popular recent methods of achieving high worst-group accuracy involve training two models. CNC (Zhang et al., 2022) first trains an ERM model to help infer pseudo-group labels by clustering output features and then adopts a standard contrastive learning approach to improve robustness. SSA (Nam et al., 2022) uses a subset of the validation samples with group annotations for training to obtain pseudo spurious attributes, and then trains a robust model by minimizing worst-group loss, the same as Group-DRO. Similarly, DFR$_{\text{Tr}}^{\text{Val}}$ (Kirichenko et al., 2022) uses validation data with group annotations for training and tuning hyperparameters, though it just requires retraining the last layer of the model. Kim et al. (2022) introduces a multi-model approach that identifies hard-to-learn samples and obtains their weights based on the consensus of member classifiers of a committee, and simultaneously trains a main classifier through knowledge distillation.

Most related to this paper are the approaches that use one model to identify minority samples and then train a second model based on the results predicted by the first model (Yaghoobzadeh et al., 2019; Utama et al., 2020). LfF (Nam et al., 2020) trains two models concurrently, where one model is intentionally biased, and the other one is debiased by reweighting the gradient of the loss according to a relative difficulty score. JTT (Liu et al., 2021) first trains an ERM model to identify minority groups in the training set (similar to EIIL (Creager et al., 2021)), and then trains a second ERM model with these selected samples to be upweighted. Hwang et al. (2022) trains an auxiliary model with generalized supervised contrastive loss to learn biased features, and then implicitly upweights of minority groups based on the mixup method (Zhang et al., 2017). Our algorithm improves over these methods by introducing trainable per-example auxiliary variables for bias amplification in the first model, as well as using continued training instead of training a separate second model.

**Improving Group Robustness without any Group Annotations.** Relatively less work has been done under the condition that no group information is provided for either training or validation. Idrissi et al. (2022); Liu et al. (2021) observed a significant drop (10% - 25%) in worst-group test accuracy if using the highest

*average* validation accuracy as the stopping criterion without any group annotations. GEORGE (Sohoni et al., 2020) generates pseudo group annotations by performing clustering in the neural network's feature space, and uses the pseudo annotations to train a robust model via distributionally robust optimization. Seo et al. (2022) clustered the pseudo-attributes based on the embedding feature of a naively trained model, and then defined a trainable factor to reweight different clusters based on their sizes and target losses. However, there is a considerable performance gap between group annotation-free methods and those that use group annotations.

The method of per-example auxiliary variables was first introduced in Hu et al. (2020) for a different motivation (learning with noisy labels). Hu et al. (2020) proved that it recovers $\ell_2$ regularization when the neural network is in the Neural Tangent Kernel (Jacot et al., 2018) regime, and provided a theoretical guarantee for the noisy label learning problem. Our result shows that this method is also applicable to improving worst-group accuracy through its bias amplification effect.

## 3  Preliminaries

We adopt the group robustness formulation for spurious correlation (Sagawa et al., 2019). Consider a classification problem where each sample consists of an input $x \in \mathcal{X}$, a label $y \in \mathcal{Y}$, and a spurious attribute $a \in \mathcal{A}$. For example, in CelebA, $\mathcal{X}$ contains images of human faces and we want to classify hair color into the labels $\mathcal{Y} = \{\text{blonde}, \text{not blonde}\}$. Hair color may be highly correlated with gender $\mathcal{A} = \{\text{male}, \text{female}\}$ which is a spurious feature that also predict the label. We say that each example belongs to a group $g = (y, a) \in \mathcal{G} = \mathcal{Y} \times \mathcal{A}$.

Let $f : \mathcal{X} \to \mathcal{Y}$ be a classifier learned from a training dataset $D = \{(x_i, y_i)\}_{i=1}^n$. We hope that $f$ does not overly rely on the spurious feature $a$ to predict the label. To this end, we evaluate the model through its *worst-group error*:

$$\text{Err}_{\text{wg}}(f) := \max_{g \in \mathcal{G}} \mathbb{E}_{x, y|g}[\mathbb{1}[f(x) \neq y]].$$

We focus on the setting where no group annotations are available in the training dataset. We consider two cases under this setting: (1) group annotations are available in a validation set solely for the purpose of hyperparameter tuning, and (2) no group annotations are available at all. We will distinguish between these cases when comparing them with existing methods.

## 4  Our Approach: Bam

---
**Algorithm 1** BAM
---

**Input:** Training dataset $D$, number of epochs $T$ in Stage 1, auxiliary coefficient $\lambda$, and upweight factor $\mu$

    **Stage 1: Bias Amplification**
    1. Optimize $R_1(\theta, B)$ equation 1 for $T$ epochs and save the model parameters $\hat{\theta}_{\text{bias}}$.
    2. Construct the error set $E$ equation 2 misclassified by $\hat{f}_{\text{bias}}(\cdot) = f_{\hat{\theta}_{\text{bias}}}(\cdot)$.
    **Stage 2: Rebalanced Training**
    3. Continue training the model starting from $\hat{\theta}_{\text{bias}}$ to optimize $R_2(\theta)$ equation 3.
    4. Apply a stopping criterion:
        • If group annotations are available for validation, stop at the highest worst-group validation accuracy;
        • If no group annotations are available, stop at the lowest validation class difference equation 4.

---

We now present BAM, a two-stage approach to improving worst-group accuracy without any group annotations at training time. In Stage 1, we train a *bias-amplified model* and select examples that this model makes mistakes on. Then, in Stage 2, we continue to train the same model while upweighting the samples selected from Stage 1.

### 4.1 Stage 1: Bias Amplification

The key intuition behind previous two-stage approaches (*e.g.* JTT) is that standard training via ERM tends to first fit easy-to-learn groups with spurious correlations, but not the other hard-to-learn groups where spurious correlations are not present. Therefore, the samples that the model misclassified in the first stage can be treated as a proxy for hard-to-learn groups and used to guide the second stage.

We design a bias-amplifying scheme in Stage 1 with the aim of identifying a higher-quality error set to guide training in Stage 2. In particular, we introduce a trainable auxiliary variable for each example and add it to the output of the network. Let $f_\theta : \mathcal{X} \to \mathbb{R}^C$ be the neural network with parameters $\theta$, where $C = |\mathcal{Y}|$ is the total number of classes. We use the following objective function in Stage 1:

$$R_1(\theta, B) = \frac{1}{n} \sum_{i=1}^{n} \ell(f_\theta(x_i) + \lambda b_i, y_i). \tag{1}$$

Here, $b_i \in \mathbb{R}^C$ is the auxiliary variable for the $i$-th example in the training set, and the collection of auxiliary variables $B = (b_1, \ldots, b_n)$ is learnable and is learned together with the network parameters $\theta$ via gradient-based optimization ($B$ is initialized to be all 0). $\ell$ is the loss function. $\lambda$ is a hyperparameter that controls the strength of the auxiliary variables: if $\lambda = 0$, this reduces to standard ERM and auxiliary variables are not used; the larger $\lambda$ is, the more we are relying on auxiliary variables to reduce the loss.

The introduction of auxiliary variables makes it more difficult for the network $f_\theta$ to learn, because the auxiliary variables can do the job of fitting the labels. We expect this effect to be more pronounced for hard-to-learn examples. For example, if in normal ERM training it takes a long time for the network $f_\theta$ to fit a hard-to-learn example $(x_i, y_i)$, after introducing the auxiliary variable $b_i$, it will be much easier to use $b_i$ to fit the label $y_i$, thus making the loss $\ell(f_\theta(x_i) + \lambda b_i, y_i)$ drop relatively faster. This will prohibit the network $f_\theta$ itself from learning this example. Such effect will be smaller for easy-to-learn examples, since the network itself can still quickly fit the labels without much reliance on the auxiliary variables. Therefore, **adding auxiliary variables amplifies the bias toward easy-to-learn examples**, making hard examples even harder to learn. We note that the trivial solution where the model achieves zero loss without actually learning will not occur with a proper choice of $\lambda$.

At the end of Stage 1, we evaluate the obtained model $\hat{f}_{\text{bias}}(\cdot) = f_{\hat{\theta}_{\text{bias}}}(\cdot)$ on the training set and identify an *error set*: (note that auxiliary variables are now removed)

$$E = \{(x_i, y_i) \colon \hat{f}_{\text{bias}}(x_i) \neq y_i\}. \tag{2}$$

### 4.2 Stage 2: Rebalanced Training

In Stage 2, we continue training the model starting from the parameters $\hat{\theta}_{\text{bias}}$ from Stage 1, using a rebalanced loss that upweights the examples in the error set $E$:

$$R_2(\theta) = \mu \sum_{(x,y) \in E} \ell(f_\theta(x), y) + \sum_{(x,y) \in D \setminus E} \ell(f_\theta(x), y), \tag{3}$$

where $\mu$ is a hyperparamter (upweight factor). For fair comparison, our implementation of upweighting follows that of JTT (Liu et al., 2021), i.e., we construct an upsampled dataset containing examples in the error set $\mu$ times and all other examples once.

We note that more complicated approaches have been proposed for Stage 2, *e.g.* Zhang et al. (2022), but we stick with the simple rebalanced training method in order to focus on the bias amplification effect in Stage 1.

### 4.3 Stopping Criterion without any Group Annotations – Class Difference

When group annotations are available in a validation set, we can simply use the *worst-group validation accuracy* as a stopping criterion and to tune hyperparameters, similar to prior approaches (Nam et al., 2020; Liu et al., 2021; Creager et al., 2021). When no group annotations are available, a naive approach is to use

the validation average accuracy as a proxy, but this results in poor worst-group accuracy (Liu et al., 2021; Idrissi et al., 2022).

We identify a simple heuristic when no group annotations are available, using *minimum class difference*, which we find to be highly effective and result in little or no loss in worst-group accuracy. For a classification problem with $C$ classes, we calculate the average of pairwise validation accuracy differences between classes as

$$\text{ClassDiff} = \frac{1}{\binom{C}{2}} \sum_{1 \leq i < j \leq C} |\text{Acc}(\text{class } i) - \text{Acc}(\text{class } j)|. \tag{4}$$

ClassDiff can be calculated on a validation set without any group annotations. The main intuition for ClassDiff is that we expect the worst-group accuracy to be (near) highest when all group accuracies have a relatively small gap between each other. Below we present a simple claim showing that ClassDiff being small is a necessary condition for having a small group accuracy difference.

**Claim 4.1** (proved in Appendix C)**.** *If the accuracies of any two groups differ by at most $\epsilon$, then* $\text{ClassDiff} \leq \epsilon$.

In all the datasets (where $C = 2$ or $3$) we experiment with, as well as for varying dataset size, class imbalance ratio, and group imbalance ratio, we observe that ClassDiff inversely correlates with worst-group accuracy (see Section 6.3) as long as ClassDiff does not fluctuate around the same value. Our results suggest that ClassDiff is a useful stopping criterion when no group annotations are available.

Our algorithm is summarized in Algorithm 1. It has three hyperparameters: $T$ (number of epochs in Stage 1), $\lambda$ (auxiliary variable coefficient), and $\mu$ (upweight factor). We provide full training details in Appendix B.

## 5 Experiments

In this section, we demonstrate the effectiveness of BAM in improving worst-group accuracy compared to prior methods that are trained without spurious attributes on standard benchmark datasets.

### 5.1 Setup

We conduct our experiments on four popular benchmark datasets containing spurious correlations. Two of them are image datasets: Waterbirds and CelebA, and the other two are NLP datasets: MultiNLI and CivilComments-WILDS. The full dataset details are in Appendix A. BAM is trained in the absence of training group annotations throughout all experiments. We obtain the main results of BAM via tuning with and without group annotations on the validation set, following Algorithm 1.

For a fair comparison, we adopt the general settings from previous methods (JTT) and stay consistent with other approaches without extensive hyperparameter tuning (batch size, learning rate, regularization strength in Stage 2, etc.). We use pretrained ResNet-50 (He et al., 2016) for image datasets, and pretrained BERT (Devlin et al., 2019) for NLP datasets. More details can be found in Appendix B.

### 5.2 Results

Tables 1 and 2 report the average and worst-group test accuracies of BAM and compare it against standard ERM and recently proposed methods under different conditions, including SUBG (Idrissi et al., 2022), JTT (Liu et al., 2021), SSA (Nam et al., 2022), CNC (Zhang et al., 2022), GEORGE (Sohoni et al., 2020), BPA (Seo et al., 2022), and Group-DRO (Sagawa et al., 2019). We tune BAM and JTT according to the highest worst-group validation accuracy (not group annotation-free) and minimum class difference (group annotation-free).

First, compared with other methods that use group annotations only for hyperparameter tuning, BAM consistently achieves higher worst-group accuracies across all datasets, with the exception of the CelebA dataset on which CNC achieves better performance. We note that CNC primarily focuses on improving Stage 2 with a more complicated contrastive learning method, while BAM uses the simple upweighting method.

Table 1: Average and worst-group test accuracies of different approaches evaluated on image datasets (Waterbird and CelebA). We run Bam and Jtt (in *) on 3 random seeds based on the highest worst-group validation accuracies and minimum class differences, respectively, and report the mean and standard deviation. Results of EIIL and Group-DRO are reported by Nam et al. (2022), and results of other approaches come from their original papers. The best worst-group accuracies under the same condition are marked in **bold**.

| Group annotation free? | Annotation only used for tuning h-params? | Method | Waterbird | | CelebA | |
|---|---|---|---|---|---|---|
| | | | Avg. | Worst-group | Avg. | Worst-group |
| No | No | SUBG (Idrissi et al., 2022) | - | $89.1_{\pm1.1}$ | - | $85.6_{\pm2.3}$ |
| | | GroupDRO (Sagawa et al., 2019) | $91.8_{\pm0.48}$ | $\mathbf{89.2}_{\pm0.18}$ | $93.1_{\pm0.21}$ | $88.5_{\pm1.16}$ |
| | | SSA (Nam et al., 2022) | $92.2_{\pm0.87}$ | $89.0_{\pm0.55}$ | $92.8_{\pm0.11}$ | $\mathbf{89.8}_{\pm1.28}$ |
| | Yes | ERM | 97.3 | 72.6 | 95.6 | 47.2 |
| | | EIIL (Creager et al., 2021) | 96.9 | 78.7 | 91.9 | 83.3 |
| | | CNC (Zhang et al., 2022) | $90.9_{\pm0.1}$ | $88.5_{\pm0.3}$ | $89.9_{\pm0.5}$ | $\mathbf{88.8}_{\pm0.9}$ |
| | | Jtt* | $89.9_{\pm0.41}$ | $86.8_{\pm1.61}$ | $91.3_{\pm0.36}$ | $78.7_{\pm1.15}$ |
| | | Bam | $91.4_{\pm0.44}$ | $\mathbf{89.2}_{\pm0.26}$ | $88.0_{\pm0.37}$ | $83.5_{\pm0.94}$ |
| Yes | - | GEORGE (Sohoni et al., 2020) | 95.7 | 76.2 | 94.8 | 52.4 |
| | | BPA (Seo et al., 2022) | - | 71.4 | - | $\mathbf{82.5}$ |
| | | Jtt + ClassDiff* | $88.5_{\pm1.47}$ | $87.1_{\pm0.24}$ | $91.8_{\pm0.76}$ | $75.4_{\pm3.28}$ |
| | | Bam + ClassDiff | $91.4_{\pm0.31}$ | $\mathbf{89.1}_{\pm0.15}$ | $88.4_{\pm2.32}$ | $\mathbf{80.1}_{\pm3.32}$ |

Table 2: Average and worst-group test accuracies of different approaches evaluated on natural language datasets (MultiNIL and CivilComments-WILDS), following the same conventions in Table 1.

| Group annotation free? | Annotation only used for tuning h-params? | Method | MultiNLI | | CivilComments-WILDS | |
|---|---|---|---|---|---|---|
| | | | Avg. | Worst-group | Avg. | Worst-group |
| No | No | SUBG (Idrissi et al., 2022) | - | $68.9_{\pm0.8}$ | - | $\mathbf{71.8}_{\pm0.4}$ |
| | | GroupDRO (Sagawa et al., 2019) | $81.4_{\pm1.40}$ | $\mathbf{76.6}_{\pm0.41}$ | $87.7_{\pm1.35}$ | $69.1_{\pm1.53}$ |
| | | SSA (Nam et al., 2022) | $79.9_{\pm0.87}$ | $76.6_{\pm0.66}$ | $88.2_{\pm1.95}$ | $69.9_{\pm2.02}$ |
| | Yes | ERM | 82.4 | 67.9 | 92.6 | 57.4 |
| | | EIIL (Creager et al., 2021) | 79.4 | 70.9 | 90.5 | 67.0 |
| | | CNC (Zhang et al., 2022) | - | - | $81.7_{\pm0.5}$ | $68.9_{\pm2.1}$ |
| | | Jtt* | $80.0_{\pm0.41}$ | $68.1_{\pm0.90}$ | $87.2_{\pm1.65}$ | $77.7_{\pm1.70}$ |
| | | Bam | $79.6_{\pm1.11}$ | $\mathbf{71.2}_{\pm1.56}$ | $88.3_{\pm0.76}$ | $\mathbf{79.3}_{\pm2.69}$ |
| Yes | - | GEORGE (Sohoni et al., 2020) | - | - | - | - |
| | | BPA (Seo et al., 2022) | - | - | - | - |
| | | Jtt + ClassDiff* | $81.2_{\pm0.56}$ | $66.5_{\pm0.56}$ | $87.2_{\pm1.65}$ | $77.7_{\pm1.70}$ |
| | | Bam + ClassDiff | $80.3_{\pm0.99}$ | $\mathbf{70.8}_{\pm1.52}$ | $88.3_{\pm0.76}$ | $\mathbf{79.3}_{\pm2.69}$ |

It is possible that the combination of CNC and Bam could lead to better results. Nevertheless, the result of Bam is promising on all other datasets, even comparable to the weakly-supervised method SSA and the fully-supervised method Group-DRO on Waterbirds and CivilComments-WILDS.

Second, if the validation group annotations are not available at all, Bam achieves the best performance on all four benchmark datasets among all the baseline methods. While BPA slightly outperforms Bam on the CelebA dataset, Bam still achieves the best overall worst-group accuracy on the image classification datasets.

Bam's improved performance in worst-group accuracy comes at the expense of a moderate drop in average accuracy. The tradeoff between average accuracy and worst-group accuracy is consistent with the observation made by Liu et al. (2021); Sagawa et al. (2019). We note that our implementation of Jtt follows directly from its published code, and we obtain a much higher performance on the CivilComments-WILDS dataset than originally reported by Liu et al. (2021).

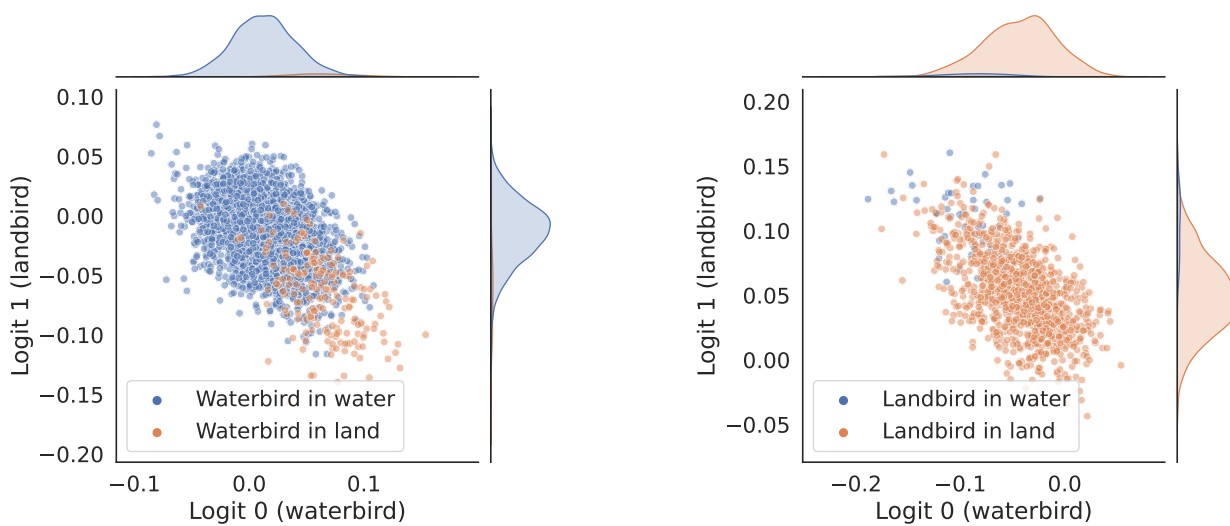

Figure 2: Distributions of the auxiliary variable w.r.t. the waterbird class (left) and landbird class (right) on the training set at $T = 20$. We use two distinct colors to illustrate the distributions of two groups in each class. The coordinates of data sample $i$ relative to the origin show the bias learned by the auxiliary variable.

## 6 Analyses and Ablations

In this section, we perform detailed analyses and ablations to better understand various components of BAM, including auxiliary variables (Section 6.2), ClassDiff (Section 6.3), sensitivity of hyperparameters $\lambda$ and $T$ (Section 6.4), and the advantage of continued training in Stage 2 (Section 6.5).

### 6.1 Additional Controlled Experiments Setup

Despite the popularity of the benchmark datasets, they have significantly different major-versus-minor class/group ratios and uncommon learning patterns (Idrissi et al., 2022). For a more rigorous analysis of our proposed auxiliary variable and classDiff, we introduce three additional controlled datasets where we adjust different class/group imbalance ratios, including Colored-MNIST, Controlled-Waterbirds, and CIFAR10-MNIST. See Appendix A for details.

### 6.2 Analysis of Auxiliary Variables

We verify the intuition behind the auxiliary variables by visualizing their learned values and comparing between majority and minority examples. Recall that we expect minority examples to rely more on auxiliary variables (see Section 4.1).

#### 6.2.1 Visualization of Auxiliary Variables

In Figure 2, we visualize the distribution of the learned auxiliary variables on each group of the original Waterbirds dataset at $T = 20$. We have two observations. First, auxiliary variables for examples in majority and minority groups have clear distinctions. Second, auxiliary variables for majority group examples are in general closer to the origin, while those in minority groups tend to have larger (positive) logit values on the ground truth class they actually belong to and have smaller (negative) logit values on the class they do not belong to. The visualization shows that the auxiliary variables help with fitting hard-to-learn samples, which corroborates the intuition described in Section 4.1. We observe the same trend for any $\lambda \in \{0.5, 5, 20, 50, 70, 100\}$ as well as any $T$ in Stage 1 in $\{20, 50, 100\}$. This supports our intuition that "adding auxiliary variables amplifies the bias toward easy-to-learn examples" in Section 4.1.

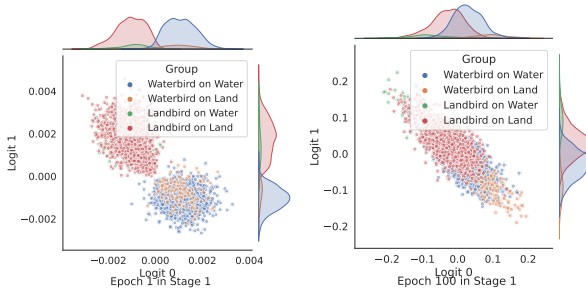

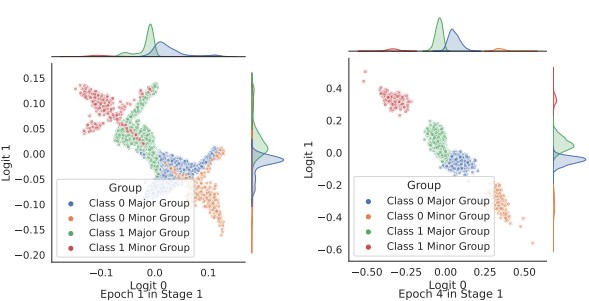

Figure 3: Epochs 1 and 100 in Stage 1 on Waterbirds. Logit 0 corresponds to the prediction on the waterbird class, and logit 1 corresponds to landbird. The group sizes are 1800, 200, 200, and 1800 in order.

Figure 4: Epochs 1 and 4 in stage 1 on Colored-MNIST. Logit 0 corresponds to the prediction on class 0, and logit 1 corresponds to class 1.

### 6.2.2 Visualization as Training Progresses

To further explore if the auxiliary variables keep such trends in different scenarios, we take a deeper look into the training progression of the auxiliary variables in our controlled experiments, where we set a variety of different imbalance sizes/ratios. In Figures 3 and 4, scatter plots show how the values of auxiliary variables change as Stage 1 proceeds. Carefully controlling all other parameters and hyperparameters, we observe clear patterns on Controlled-Waterbirds and Colored-MNIST: First, as $T$ grows, the logits become larger and increasingly influence the model predictions. Second, minority and majority groups are more separated from each other as training progresses, resulting in the bias amplification effect. Appendix D illustrates the values of $b$ at more intermediate epochs $T$ and clearly shows the trends.

### 6.3 Results of ClassDiff

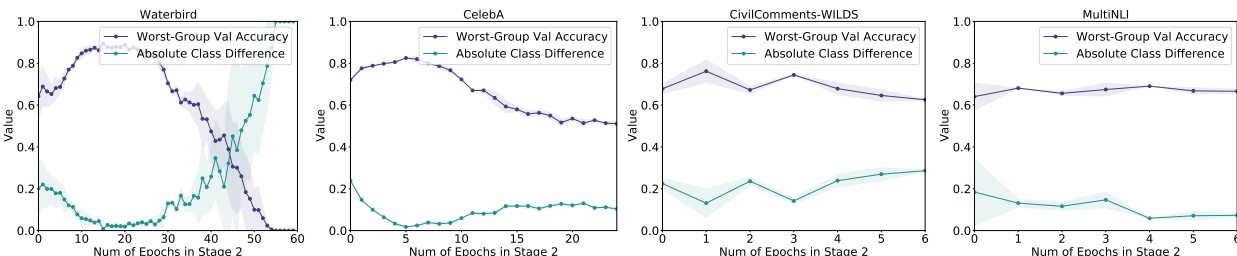

Figure 5: Relation between absolute valdiation class difference and worst-group validation accuracy in Stage 2 on Waterbirds, CelebA, CivilComments-WILDS, and MultiNLI. It can be observed that minimizing absolute validation class difference is roughly equivalent to maximizing worst-group accuracy in the validation set. Each dataset uses the same hyperparameters as employed in Table 1. Each line represents the value averaged over 3 different seeds and the shade represents the standard deviation.

We perform extensive experiments on the four benchmark datasets, as well as on our controlled datasets by altering class and group imbalance ratios. We find that ClassDiff works well on every single experiment, which shows the effectiveness and robustness of ClassDiff across different settings.

**Real-World Benchmark Datasets** Figure 5 plots the trend of the absolute class difference and the worst-group accuracy on the validation set in Stage 2 for the Waterbirds, CelebA, MultiNLI, and CiviComments. Clearly, there is a strong inverse relationship between the absolute class difference and worst-group accuracy in the validation set on all four datasets, which justifies the use of class difference as a stopping criterion.

**Controlled Datasets** On our controlled datasets (Colored-MNIST, CIFAR10-MNIST, and Controlled-Waterbirds), we vary the dataset sizes, class size ratios, and group size ratios. In Appendix E, we show

the performance of ClassDiff in several randomly selected settings (we explain our procedure of selecting settings that do not involve cherry-picking). In *every single setting* we have tested, we observe a similar negative correlation as in Figure 5. These findings suggest that ClassDiff could be generally effective when no group annotation is available, as it is robust across different datasets, varying dataset sizes, and imbalanced characteristics.

### 6.4 Sensitivity of Hyperparameters

We find that BAM is robust to the choice of $\lambda$, as Table 3 below presents the best worst-group accuracies for a wide range of $\lambda$ on Waterbirds. Clearly, BAM demonstrates resilience against the choice of $\lambda$. We also find that there is a noticeable performance drop when $\lambda = 0$ compared with $\lambda > 0$, further demonstrating its efficacy.

Table 3: Sensitivity of $\lambda$

| $\lambda$ | 0 | 0.5 | 10 | 30 | 35 | 40 | 50 | 70 | 100 |
|---|---|---|---|---|---|---|---|---|---|
| Test acc | 87.1 | 89.9 | 89.8 | 89.7 | 90.2 | 89.5 | 89.5 | 88.8 | 88.6 |

Table 4: Robustness of worst group test accuracy with different $T$'s

| $\lambda$ | 0 | 0 | 0 | 5 | 5 | 5 | 20 | 20 | 20 | 30 | 30 | 30 | 40 | 40 | 40 | 50 | 50 | 50 | 70 | 70 | 70 |
|---|---|---|---|---|---|---|---|---|---|---|---|---|---|---|---|---|---|---|---|---|---|
| $T$ | 80 | 100 | 120 | 80 | 100 | 120 | 80 | 100 | 120 | 80 | 100 | 120 | 80 | 100 | 120 | 80 | 100 | 120 | 80 | 100 | 120 |
| Test acc | 82.2 | 75.2 | 68.9 | 83.5 | 77.4 | 75.7 | 84.7 | 81.8 | 77.2 | 86.0 | 86.9 | 81.8 | 87.1 | 86.7 | 85.3 | 88.6 | 88.2 | 86.9 | 88.8 | 88.8 | 88.5 |

Furthermore, using a larger $\lambda$ can eliminate the need to carefully tune the epoch number $T$ in Stage 1. This resolves a major drawback of previous methods such as JTT, whose performance is sensitive with respect to the choice of $T$. Table 4 shows that even trained until full convergence in Stage 1, an appropriate choice of $\lambda$ can still guarantee a competitive performance. In addition, as $\mu$ increases, there are clear performance boosts with larger $T$s'. This strong correlation shows the effect of $\mu$.

Table 5: Sensitivity of $\mu$

| $\mu$ | 20 | 30 | 40 | 50 | 60 | 70 | 80 | 90 | 100 | 110 | 120 | 130 | 140 | 150 | 160 | 170 | 180 |
|---|---|---|---|---|---|---|---|---|---|---|---|---|---|---|---|---|---|
| Test acc | 80.1 | 85.8 | 83.8 | 85.6 | 87.6 | 88.8 | 88.7 | 88.9 | 88.9 | 89.6 | 89.0 | 89.7 | 89.2 | 89.0 | 89.2 | 90.1 | 88.4 |

Finally, Table 5 examine the sensitivity of $\mu$. We find that a large range of $\mu$ (70-180) can guarantee a very competitive performance. This result further illustrates that BAM is not very sensitive to the choice of hyperparameters.

### 6.5 Ablation Studies on *One-M* vs. *Two-M*

We conduct ablation studies on the Waterbirds dataset to test the effectiveness of our proposed method of continued training in Stage 2. For consistent terminology, we define the approach that loads the model from Stage 1 and continues training the same model in Stage 2 as *One-M*. We define the approach that trains a separate ERM model in Stage 2 as *Two-M*. For a fair comparison, we employ the same hyperparameter settings throughout this section. We use the same stopping criterion here (highest worst-group validation accuracy). We tune over $T = \{10, 15, 20, 25, 30, 40, 50, 60, 80, 100, 120\}$ and $\mu = \{50, 100, 140\}$ for each setting, and fix $\lambda = 50$. Each experiment is replicated thrice with varying seeds to counteract randomness, subsequently averaging results.

Figure 6 compares the performance between *One-M* and *Two-M* over a wide range of Stage 1 epochs $T$. Notably, *One-M* and *Two-M* share the same error sets in each $T$. The result suggests that *One-M* BAM outperforms *Two-M* BAM in every single Stage 1 epoch $T$, further verifying our intuition that the biased knowledge accumulated from Stage 1 helps with improving the final robustness. Additionally, as explained in Section 6.4 and also shown here in Figure 6, BAM demonstrates a stable performance, regardless of the choice of $T$'s. By stark contrast, JTT's performance falters with escalating $T$, as documented in Figure 3 of Liu et al. (2021).

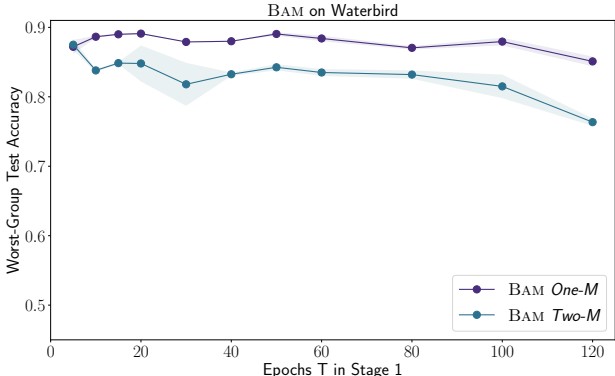

Figure 6: Comparison of Bam's performance between training *One-M* and *Two-M*. We find that *One-M* consistently achieves better worst-group test accuracies than *Two-M* for Bam over all hyperparameter combinations and Bam *One-M* demonstrates a stable performance regardless of the choice of $T$.

## 7 Conclusion

In this work, we introduced Bam, a method that can effectively improve the worst-group accuracy across different NLP and CV benchmarks. Further experiments suggest the effectiveness of the bias amplification scheme of the auxiliary variable and ClassDiff through comprehensive analysis and under various experiment settings. Future work can further leverage the idea of Bam and apply the auxiliary variables to other applications, such as continual learning and curriculum learning. Conducting theoretical analysis of the bias amplification scheme can also be of great interest.

## 8 Ethical Statement

Bam is designed to effectively enhance worst-group robustness, and it represents a step towards addressing the marginalization of under-represented groups in the training data. We believe that a training procedure without learning spurious correlation is essential for the responsible development of deep learning tools and models. In addition, while we evaluate Bam on four well recognized benchmark datasets, we acknowledge that these may not fully encapsulate the complexity and biases present in real-world data. This limitation underscores the need for more comprehensive and diverse datasets in future works. Furthermore, the bias-amplified model trained in Stage 1 should not be deployed for real-world deployment and the model must be continued trained in Stage 2 to correct the bias.

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

# A    Dataset details

**Waterbirds** (Wah et al., 2011; Sagawa et al., 2019): Waterbirds is an image dataset consisting of two types of birds on different backgrounds. It contains images of birds directly cut and pasted on real-life images of different landscapes. We aim at classifying $\mathcal{Y} = \{$waterbird, landbird$\}$, which is spuriously correlated with the background $\mathcal{A} = \{$water, land$\}$. The train/validation/test splits is followed from Sagawa et al. (2019).

**CelebA** (Liu et al., 2015): CelebA is an image dataset consisting of celebrities with two types of hair color and different genders. we aim at classifying celebrities' hair color $\mathcal{Y} = \{$blond, not blond$\}$, which is spuriously correlated with the gender $\mathcal{A} = \{$male, female$\}$. The train/validation/test splits is followed from Sagawa et al. (2019).

**MultiNLI** (Williams et al., 2018): MultiNLI is a natural language processing dataset, where each sample consists of two sentences and the second sentence is entailed by, neutral with, or contradicts the first sentence, with the presence or absence of negation words. We aim at classifying the sentence relationship $\mathcal{Y} = \{$entailment, neutral, contradiction$\}$, which is spuriously correlated with the negation words $\mathcal{A} = \{$negation, no negation$\}$. The train/validation/test splits is followed from Sagawa et al. (2019).

**CivilComments-WILDS** (Borkan et al., 2019; Koh et al., 2021): CivilComments-WILDS is a natural language processing dataset consisting of online comment that is toxic or non-toxic, with or without the mentions of certain demographic identities (male, female, White, Black, LGBTQ, Muslim, Christian, and other religions). We aim at classifying whether the sentence is toxic $\mathcal{Y} = \{$toxic, non-toxic$\}$, which is spuriously correlated with the mentions of (demographic) identities $\mathcal{A} = \{$identity, no identity$\}$. The train/validation/test splits is followed from Koh et al. (2021).

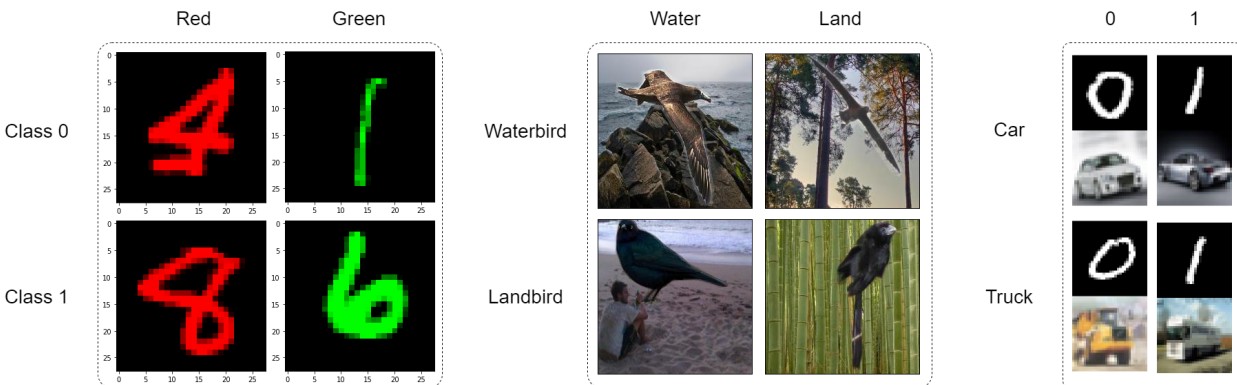

Figure 7: Examples of each group in Colored-MNIST, Controlled-Waterbirds, and CIFAR10-MNIST.

**Colored-MNIST** (Arjovsky et al., 2019; Cireşan et al., 2011): Colored-MNIST is an image dataset that colors each grayscale MNIST images either green or red and make it spuriously correlates with the digit values. We aim at classifying binary class labels $\mathcal{Y} = \{$class 0 (digits 0-4), class 1 (digits 5-9)$\}$ without relying on the color $\mathcal{A} = \{$red, green$\}$. Examples are shown in Figure 7. We shuffle the original data and generate dataset splits with train/valudation/test sizes $= 0.7/0.15/0.15$. In Figure 4, we set the total dataset size to be 10000, class imbalance ratio 1.0, and the group imbalance ratio within each class to be 0.1.

**Controlled-Waterbirds** (Wah et al., 2011; Sagawa et al., 2019): We manually construct this dataset from the original Waterbirds benchmark. We follow the previous definition that sets Waterrbird-Water and Landbird-Land as majority groups. We regenerate the dataset splits as train/valudation/test sizes $= 0.7/0.15/0.15$. Examples are shown in Figure 7. To show how the ClassDiff performs in various settings in Appendix E, we consider the following three circumstances: (1) We vary the group imbalance ratios, while the two classes are balanced. Due to the original dataset size limits, we fix the size of two majority groups to 1800, and vary the minority group sizes in $\{200, 400, 600, 800\}$. (2) The two classes are imbalanced. We fix sizes of the two groups in the smaller class as 200 and 1400, and choose the class imbalance ratio waterbird landbird

in $\{1.0, 2.0, 3.0\}$. (3) Both group and class sizes are imbalanced. We consider a random setting of four group sizes, where Waterbird-Water : Waterbird-Land : Landbird-Water : Landbird-Land = 1800 : 600 : 200 : 3600.

**CIFAR10-MNIST** (Krizhevsky et al., 2009; Cireşan et al., 2011): We manually construct this dataset by concatenating MNIST images onto CIFAR-10 images. We aim at classifying binary class labels $\mathcal{Y} = \{car, truck\}$, which is spuriously correlated with the MNIST digits $\mathcal{A} = \{0, 1\}$. We shuffle the original data and generate dataset splits with train/valudation/test sizes = 0.7/0.15/0.15. Examples are shown in Figure 7. To show how the ClassDiff performs in various settings in Appendix E, we consider the following two circumstances: (1) We vary the group imbalance ratios, while the two classes are balanced. Due to the original dataset size limits, we fix the size of two majority groups to 4000, and vary the minority group sizes in $\{400, 600, 800\}$. (2) The two classes are imbalanced. We fix sizes of the two groups in the larger class as 800 and 4000, and choose the class imbalance ratio car truck in $\{4/3, 4/2, 4/1\}$.

## B  Training details

In this section, we provide details about the model selection and the hyperparameter tuning for different datasets. As claimed in the main text, in order to make a fair comparison with previous methods, we use the same pretrained models. Namely, we use ResNet-50 pretrained from Image-net weights for Waterbirds and CelebA, and pretrained BERT for MultiNLI and CivilComments. We use the `Pytorch` implementation for ResNet50 and the `HuggingFace` implementation for BERT. We tune Bam and Jtt according to class difference and worst-group accuracies in the validation set in Stage 2.

Table 6: Hyperparameters tuned over 4 datasets.

| Dataset | Auxiliary coefficient ($\lambda$) | #Epochs in Stage 1 ($T$) | Upweight factor ($\mu$) |
|---|---|---|---|
| Waterbirds | $\{0.5, 5, 50\}$ | $\{10, 15, 20\}$ | $\{50, 100, 140\}$ |
| CelebA | $\{0.5, 5, 50\}$ | $\{1, 2\}$ | $\{50, 70, 100\}$ |
| MultiNLI | $\{0.5, 5, 50\}$ | $\{1, 2\}$ | $\{4, 5, 6\}$ |
| CivilComments | $\{0.5, 5, 50\}$ | $\{1, 2\}$ | $\{4, 5, 6\}$ |

In general, our setting follows closely from Liu et al. (2021), with some minor discrepancies. For the major hyperparameters, We tuned over $\lambda = \{0.5, 5, 50\}$, $T = \{1, 2, 10, 15, 60\}$ and $\mu = \{4, 5, 6, 50, 70, 100, 140\}$ for Bam. We note that Bam is fairly insensitive with the choice of $\lambda$ as mentioned in Section 6.4. We tune over $T = \{1, 2\}$ and $\mu = \{4, 5, 6, 50, 70, 100, 140\}$ for Jtt for fair comparisons. We tuned the major hyperparameters according to Table 6. More details are provided below:

**Waterbirds**  We use the learning rate 1e-5 and batch size 64 for two stages of training. We use the stochastic gradient descent (SGD) optimizer with momentum 0.9 throughout the training process. We use $\ell_2$ regularization 1 for Stage 2 *rebalanced training*. We apply the same above setting for both Jtt and Bam. Notably, as illustrated by Figure 5, when tuned for the minimum absolute validation class difference, the curve may fluctuate abnormally after the first 30 epochs and it is clear that the model is not learning anything useful. We tackle this problem by smoothing out abrupt changes in class difference (neglect the result if the difference between consecutive class differences is greater than 10%). The best result for Jtt occurs when $T = 60$ and $\mu = 140$. The best result for Bam occurs when $T = 20$ and $\mu = 140$. We train for a total of 360 epochs

**CelebA**  We use the learning rate 1e-5 and batch size 128 for two stages of training. We use the stochastic gradient descent (SGD) optimizer with momentum 0.9 throughout the training process. We use $\ell_2$ regularization 0.1 for Stage 2 *rebalanced training*. We apply the same above setting for both Jtt and Bam. The best result for Jtt occurs when $T = 1$ and $\mu = 70$. The best result for Bam occurs when $T = 1$ and $\mu = 50$. We train for a total of 60 epochs.

**MultiNLI**   We use batch size 32 for two stages of training. We apply an initial learning rate of 2e-5 for Stage 1 and 1e-5 for Stage 2. We use the SGD optimizer without clipping for Stage 1 and the AdamW optimizer with clipping for Stage 2. We use $\ell_2$ regularization 0.1 for Stage 2 *rebalanced training*. We apply the same above setting for both JTT and BAM. The best result for JTT occurs when $T = 2$ and $\mu = 4$. The best result for BAM occurs when $T = 2$ and $\mu = 6$. We train for a total of 10 epochs.

**CivilComments-WILDS**   We use batch size 16 for two stages of training. We apply an initial learning rate of 2e-5 for Stage 1 and 1e-5 for Stage 2. We use the SGD optimizer without clipping for Stage 1 and the AdamW optimizer with clipping for Stage 2. We use $\ell_2$ regularization 0.01 for Stage 2 *rebalanced training*. We apply the same above setting for both JTT and BAM. The best result for JTT occurs when $T = 1$ and $\mu = 4$. The best result for BAM occurs when $T = 1$ and $\mu = 4$. We train for a total of 10 epochs.

**Colored-MNIST**   We use batch size 64 and learning rate 1e-4 for two stages of training. We use the stochastic gradient descent (SGD) optimizer with momentum 0.9 throughout the training process. We use a simple convolution network with two convolutional layers followed by a fully connected layer. As a toy dataset, the dataset is only used for visualizations. In Figure 4, we use $\lambda = 20$.

**Controlled-Waterbirds**   We fix $\lambda = 50$ and $T = 100$. For circumstances (1), we test different $\mu \in \{70, 100, 120\}$. For circumstance (2), we test different $\mu \in \{10, 50, 70, 100, 120, 150\}$. For ciscumstance (3), we test different $\mu \in \{10, 50, 70, 100, 120, 150\}$. All other hyperparameters and settings follow what is used in the original Waterbirds dataset.

**CIFAR10-MNIST**   We use the learning rate 1e-5 and batch size 64 for two stages of training. We use the stochastic gradient descent (SGD) optimizer with momentum 0.9 throughout the training process. We use $\ell_2$ regularization 1 for Stage 2 *rebalanced training*. We use pretrained ResNet-18 as the training model. We fix $\lambda = 50$, and test different $T \in \{5, 10, 25\}$ and $\mu \in \{5, 10, 20, 50\}$ for circumstances (1) and (2).

## C   Proof of Claim 4.1

*Proof of Claim 4.1.* Suppose we have $C$ classes, and each class $i$ contains two groups $i(a)$ and $i(b)$. We have that all group accuracies (*i.e.* $1(a), 1(b), 2(a), 2(b), \ldots, C(a), C(b)$) are within $\epsilon$ of each other, then

$$
\begin{aligned}
\text{ClassDiff} &= \frac{1}{\binom{C}{2}} \sum_{1 \leq i < j \leq C} |\text{Acc}(\text{class } i) - \text{Acc}(\text{class } j)| \\
&= \frac{1}{\binom{C}{2}} \sum_{1 \leq i < j \leq C} |\alpha_i \text{Acc}(ia) + (1 - \alpha_i)\text{Acc}(ib) - \text{Acc}(ja) - (1 - \alpha_j)\text{Acc}(jb)| \\
&\leq \frac{1}{\binom{C}{2}} \sum_{1 \leq i < j \leq C} \epsilon = \epsilon,
\end{aligned}
$$

where $\alpha_i$ refers to the group ratio of group $a$ in class $i$.  □

## D   Further Analysis on Auxiliary Variables

Figure 8 and Figure 9 show the auxiliary variable logits with different $T$'s in Stage 1, and clearly illustrate the trends of changes.

## E   Class Difference Plots and Explanations

We conducted additional experiments on two datasets to verify the robustness of BAM by changing the relative group sizes and class sizes: Controlled-Waterbirds and CIFAR10-MNIST. Considering the circumstances listed in Appendix A and following the training details in Appendix B, we conduct the experiments, and randomly show some of the ClassDiff plots in Figure 13.

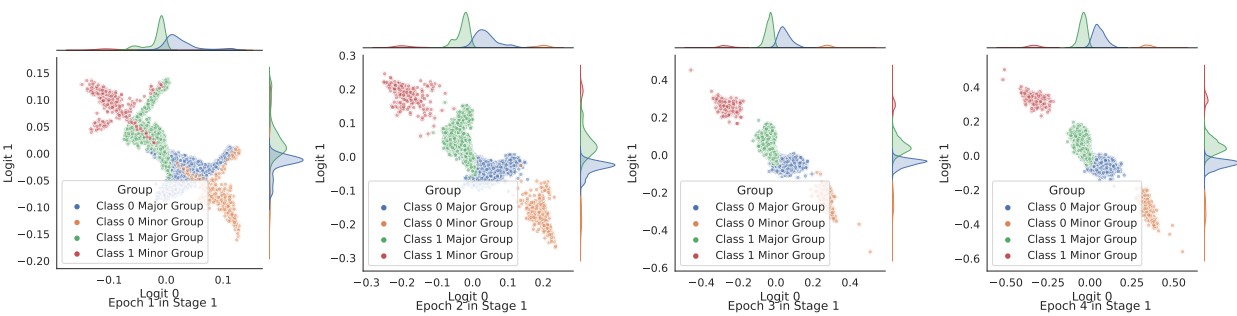

Figure 8: Epoch 1 to 4 in stage 1 on a toy dataset, Colored-MNIST. Logit 0 corresponds to the prediction on class 0, and logit 1 corresponds to class 1.

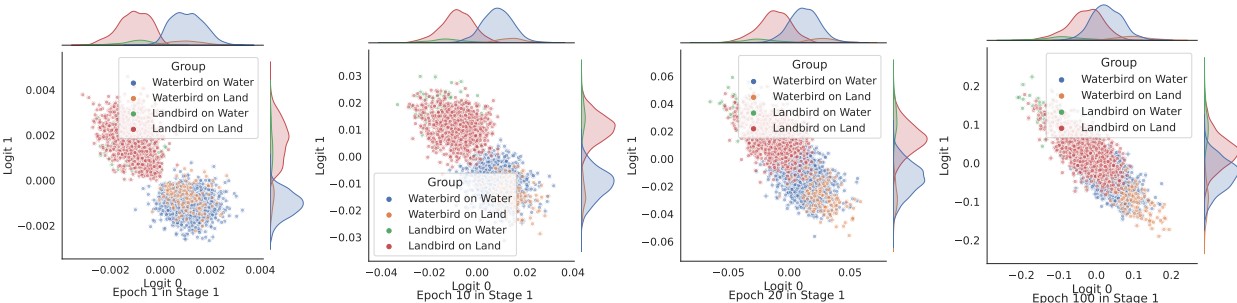

Figure 9: Epoch 1 to 100 in Stage 1 on Waterbirds. Logit 0 corresponds to the prediction on the waterbird class, and logit 1 corresponds to landbird. The group sizes are 1800, 200, 200, 1800 in order.

To avoid charry picking, we first classify each experiment into one of the settings in Appendix A. The number of experiments are $12, 18, 6, 36, 36$ for each scenario in order. Then, we sort the following parameters and hyperparameters by the order of importance: dataset parameter values shown in each setting in the dataset details $> \lambda > T > \mu$. Finally, we set random seed to 0 and use the function `NumPy.random.choice` to select 5 independent runs for each circumstance.

The following figures illustrate that both ClassDiff and BAM are robust to varying group and class ratios on the manually generated datasets. In addition, we observed on every single experiment of both these two new established datasets that the class difference is negatively correlated with highest robust test accuracy, regardless of their group sizes and class sizes.

## F    Smaller Validation Set

Table 7 shows the worst-group accuracies on Waterbirds with varying sizes of group-annotated validation set.

Table 7: Worst-group accuracy on Waterbirds with varying size of validation set with group annotations. BAM maintains high worst-group test accuracies even when tuned with very few numbers of group-annotated validation set. However, we note that with a fewer size of validation set with group annotations, the performance is actually worse than when tuned for a full-size validation set without any group annotations.

| Size of Annotated Validation Set | 100% | 20% | 10% | 5% |
|---|---|---|---|---|
| Worst-Group Acc.(%) | 89.8 | 89.1 | 88.4 | 86.2 |

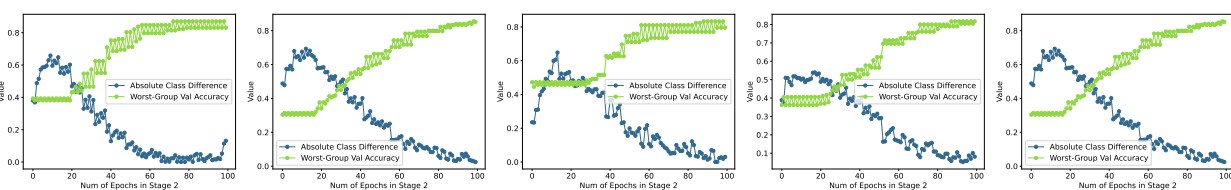

Figure 10: 5 random experiments of the circumstance (1) on the Controlled-Waterbirds dataset.

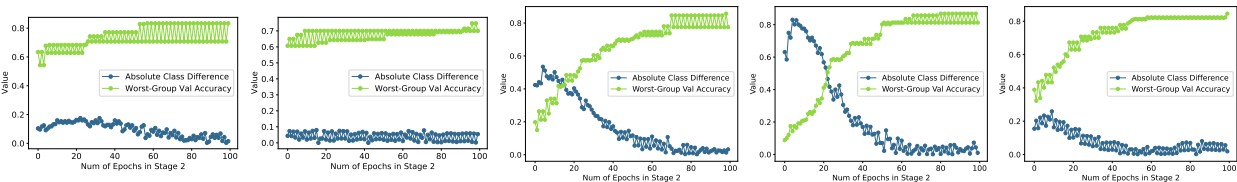

Figure 11: 5 random experiments of the circumstance (2) on the Controlled-Waterbirds dataset.

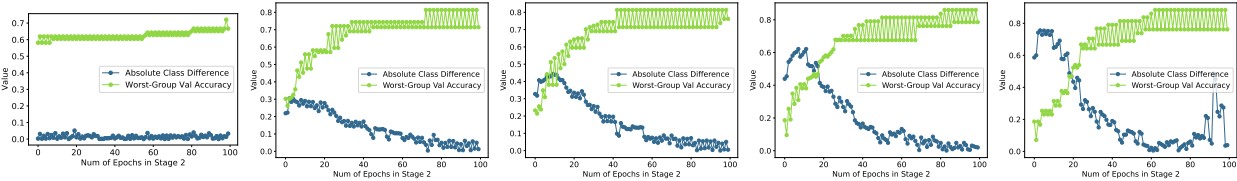

Figure 12: 5 random experiments of the circumstance (3) on the Controlled-Waterbirds dataset.

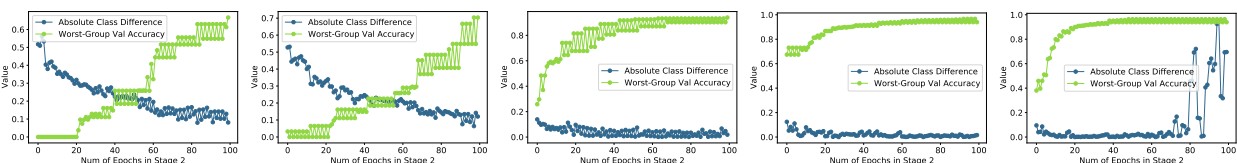

Figure 13: 5 random experiments on the CIFAR10-MNIST dataset.

# G    Supplementary Figures

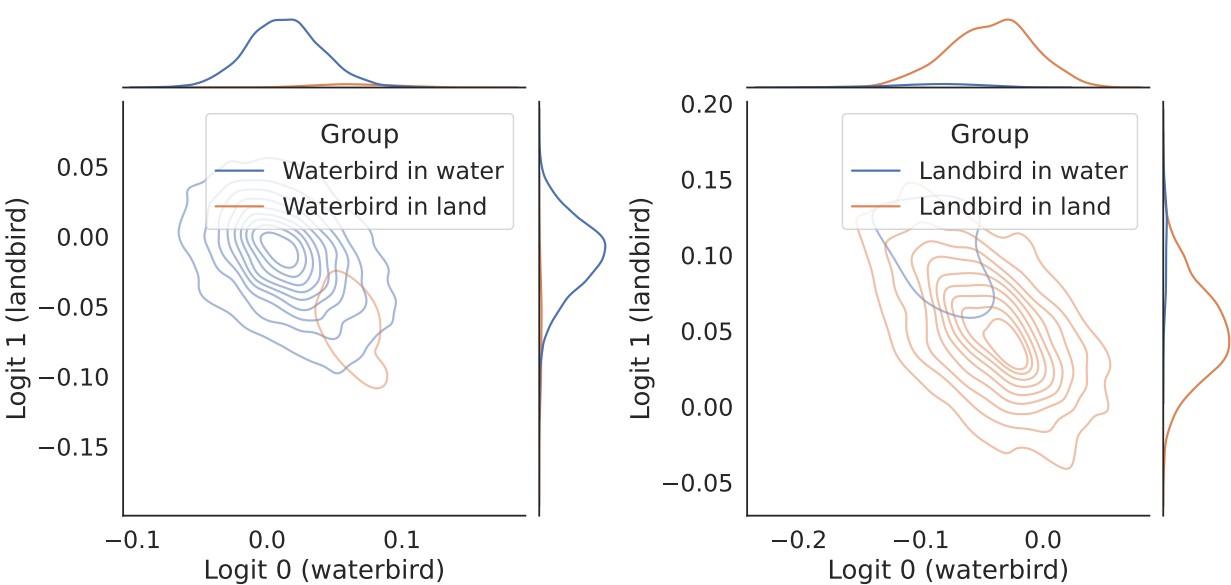

Figure 14: Corresponding KDE plot of Figure 2.

