# OpenReview forum: "Bias Amplification Enhances Minority Group Performance"
_TMLR — Accepted by TMLR_

### Review · Reviewer_PXZp · 2023-10-15

**Summary Of Contributions:**

The article introduces a method called Bias Amplification (Bam) that aims to improve worst-group accuracy in machine learning models. The method consists of two stages: bias amplification and rebalanced training. The method can remove the need for group annotations by using a simple stopping criterion based on minimum class accuracy difference.

**Audience:**

Yes

**Broader Impact Concerns:**

1.	Do the datasets used for the experiment (e.g., Waterbirds, CelebA, MultiNLI, and CivilComments-WILDS) truly reflect real-world distributions and biases? What are the potential risks if the model is applied in the real world?

2.	The experiment mentions an improvement in the accuracy of "worst-group". Have the authors considered whether such improvements might exacerbate the marginalization of niche groups in some cases?

3.	Bam has been compared to CNC methods using contrast learning in some experiments. Considering that contrast learning may amplify the importance of certain features, how might this affect the model's performance in terms of diversity or inclusion?
4.	Are there potential risks for the choice of hyperparameters such as λ and T, such as overfitting or instability due to over-tuning in real-world applications?

5.	Bam uses auxiliary variables to aid in training. Are these auxiliary variables likely to lead to some unforeseen bias or misrepresentation in practical applications?

**Claims And Evidence:**

Yes

**Requested Changes:**

1.	What are the theoretical foundations behind the Bias Amplification (Bam) method? Could you explain the process of bias amplification and rebalanced training in more detail?

2.	How does the Bam method address the issue of covariate shifts and distributional differences between the source and target data?

3.	How does the Bam method handle cases where the spurious attributes are not known or are difficult to identify?

4.	Could you explain the process of upweighting misclassified examples in the rebalanced training stage?

5.	How does the Bam method handle cases where the training data is imbalanced in terms of both group and class sizes?

6.	The paper starts with the premise that amplifying bias in the first model can improve final group robustness. How did you determine that this approach is more effective than directly trying to reduce the bias?
7.	How does the proposed Bam algorithm differ significantly from other two-stage algorithms or existing methods that aim at bias mitigation?

8.	How does Bam ensure that it isn't simply overfitting to the "low class accuracy difference" when it's used as a stopping criterion in the absence of group annotations?

9.	For the rebalanced training in Stage 2, how does the choice of upweighting factor µ impact the model? Is there a risk of over-emphasizing the error set from Stage 1?

10.	Paper mentioned a tradeoff between average accuracy and worst-group accuracy. Could you provide a more detailed analysis on why this tradeoff occurs and its implications?

11.	What are the benefits of introducing controlled datasets like Colored-MNIST, Controlled-Waterbirds, and CIFAR10-MNIST over the standard benchmarks?

12.	Why is Bam's performance, unlike Jtt's, unaffected by full convergence in Stage 1?

13.	How does Bam's computational cost and training time compare with the other mentioned methods?

14.	Effect of Imbalance: How does the class or group imbalance in datasets affect the performance of Bam?

**Strengths And Weaknesses:**

Strengths: 1. The novel method Bam improves worst-group accuracy in machine learning models and does not require group annotations for training data.
2. The article provides comprehensive experimental results on various benchmark datasets.
3. The use of Grad-CAM visualization helps illustrate the impact of bias amplification and rebalanced training stages.

Weaknesses:1.The theoretical basis of the bias amplification scheme can be further explored and analyzed.
2. The BAM model is under-motivated and the article should provide more details about the experimental setup and a detailed analysis of the model to present a more argumentative summary.

---

> ### Author Response · Authors · 2023-11-29
> **Response to Reviewer PXZp (Part I)**
>
> Thank you so much for your early review and great clarification questions. We provide our explanations below:
>
> **Q1.1 Intuition and Theoretical Basis of Bam**
>
> When trained on datasets whose class labels are correlated with spurious attributes, prior works (JTT, CNC, etc.) empirically showed that neural networks initially fit mostly majority group examples and do not fit minority group examples until later in training. Therefore, algorithms like JTT and CNC first use ERM to identify the minority groups by looking at samples where the model misclassifies; then in the second stage, these identified examples are upweighted. Our paper introduces Bam to improve the first stage by making harder examples even harder to learn so that a better set of minority group examples can be identified.
>
> In particular, our method of adding per-example auxiliary variables is inspired by Hu et al. (2020) which provided a theoretical guarantee for such a method in the setting of learning with noisy labels. Our intuition is similar to that of Hu et al. (2020), which is that auxiliary variables play a larger role in hard-to-learn examples, making them even harder to learn. In Section 6, we did substantial experiments to show that the auxiliary variables indeed show a clear distinction between majority groups and minority groups (Figures 3-4), which aligns well with our intuition.
>
> **Q1.2 Algorithm details explanation and intuition**
>
> In Stage 1, we train the neural network model $f(\theta)$ together with the auxiliary variable $B$ (Equation 1). In Stage 2, we upweight the training examples that are misclassified by the network in the previous stage, and continue training the same model. Notice that $B$ is removed in Stage 2. The motivation of such design is that it is expected that the auxiliary variable self-identifies the hard-to-learn samples **without knowing** them in advance, and gives those samples more importance during the second stage.
>
> To provide further explanations, we answer three questions here and provide additional details:
> 1.  What does it mean by bias-amplification?
> 2.  Why is the bias-amplification scheme beneficial for improving the group robustness?
> 3.  How does Bam achieve bias-amplification?
>
> For (1), by bias amplification we mean the model is more biased towards easy-to-learn examples (learns their spurious features) and makes more mistakes on hard-to-learn examples. On the Waterbird dataset that is widely used in previous work, the model in Bam makes more mistakes on the minority samples than JTT at the end of Stage 1, presented below:
>
> | Group Size | Group Description   | JTT | Bam |
> |------------|---------------------|------|------|
> | 3498       | Waterbird in Water  | 7    | 10   |
> | 184        | Waterbird in Land   | 65   | 80   |
> | 56         | Landbird in Water   | 41   | 42   |
> | 1057       | Landbird in Land    | 116  | 90   |
>
> In the table above, we can see that Bam makes more mistakes in the minority samples (Waterbird in Land and Landbird in Water) and fewer mistakes in the majority samples (Waterbird in water and Landbird in land) than JTT. A better-quality error set (to be upweighted in the second stage) should better help the model to improve its final robustness, and such intuition is confirmed in the consistently improved performance shown in Table 1 and Table 2 in our paper. This answers the question (2).
>
> To answer (3), we set up the whole analysis section (Section 6) to further understand why our bias amplification scheme works. Sections 6.1 and 6.2 provide convincing figures that the auxiliary variables learn to differentiate majority and minority samples and the difference gets clearer as training progresses. We can see that the model relies more heavily on the auxiliary variables when making predictions for the minority samples and thus is more biased toward the majority samples.
>
> **Q2 How does the Bam method address the issue of covariate shifts and distributional differences between the source and target data?**
>
> We study the worst-group robustness question which can be viewed as a variant of covariate shift, as the source data contain a mixture of groups and the target data are the group with the worst accuracy. It would be interesting to study other distribution shift settings in future work.
>
> **Q3 How does the Bam method handle cases where the spurious attributes are not known or are difficult to identify?**
>
> If we correctly understand the question, it should already be answered in **Q1** above. Specifically, our proposed Bam method identifies hard-to-learn examples as error sets at the end of Stage 1 and upweights them in Stage 2. This process does not require any explicit knowledge of the spurious attributes. Note that Bam identifies hard-to-learn examples **automatically**, which turn out to be well correlated with minority-group examples.

---

> ### Author Response · Authors · 2023-11-29
> **Response to Reviewer PXZp (Part II)**
>
> **Q4 Could you explain the process of upweighting misclassified examples in the rebalanced training stage?**
>
> As explained in Equation 3, we up-sample those misclassified examples in the error set $\mu$ times and all other examples once. We then continue training the model on this upsampled dataset. Note that the term "upweighting" is used following the previous work (JTT) and it means up-sampling.
>
> **Q5 How does the Bam method handle cases where the training data is imbalanced in terms of both group and class sizes?**
>
> On all the four real datasets that we tested on, both class and group imbalances exist. We also did additional controlled experiments by varying the class and group imbalance ratios in Waterbirds (which we call Controlled-Waterbirds; see Appendix A for details). We found that Bam is effective on all of them.
>
> **Q6 The paper starts with the premise that amplifying bias in the first model can improve final group robustness. How did you determine that this approach is more effective than directly trying to reduce the bias?**
>
> We are unsure what it means by "directly reduce the bias." We explain the concept and the benefit of bias amplification in **Q1.2** (hopefully that already answered your questions). Note that we assume no access to group information during training. The two-stage training framework relies on the quality of the error set identified in Stage 1, and a bias-amplified model from Stage 1 will produce a higher-quality error set (see explanation in **Q1** above). We are happy to provide a more detailed response if you have any further questions.
>
> **Q7 How does the proposed Bam algorithm differ significantly from other two-stage algorithms or existing methods that aim at bias mitigation?**
>
> As explained in Section 2, our method is sufficiently different from other two-stage algorithms (CNC, JTT, etc.) due to the use of trainable auxiliary variables and changing from Two-M to One-M. We highlighted the effectiveness of these two strategies through detailed analyses and ablation studies.
>
> **Q8 How does Bam ensure that it isn't simply overfitting to the "low class accuracy difference" when it's used as a stopping criterion in the absence of group annotations?**
>
> No group annotations being available is still a largely under-explored setting due to its difficulty, though it has strong motivations in real-world applications. In this case, unlike usual classification problems for which we tune hyperparameters on validation sets to avoid overfitting, we do not have access to group information at all. Instead, we can only use data with class labels to somehow infer group information. Therefore, ClassDiff is designed to be a simple **heuristic** to identify when to stop in the second stage. To ensure Bam isn't merely overfitting, we conducted extensive experiments using both real-world datasets and controlled datasets where we are able to carefully test the effectiveness of our proposed stopping criterion under various degrees of group and class imbalances. We found that in all cases that we experimented with, worst-group accuracy is negatively correlated with class difference, and Bam consistently achieves high worst-group accuracy using the low class accuracy difference heuristics.
>
> **Q9 For the rebalanced training in Stage 2, how does the choice of upweighting factor µ impact the model? Is there a risk of over-emphasizing the error set from Stage 1?**
>
> The upweighting factor $\mu$ is a tunable hyperparameter as in previous works. There is indeed some performance degradation due to over-emphasizing if $\mu$ is too large. However, as long as the identified error set correlates well with the minority group examples (as supported and visualized in Figures 2, 3, 4, 8, 9, and 14 for Bam), over-emphasizing would not be a significant problem. In addition, our updated Table 5 in Section 6.4 shows that Bam achieves competitive performance for a wide range of $\mu$'s.

---

> ### Author Response · Authors · 2023-11-29
> **Response to Reviewer PXZp (Part III)**
>
> **Q10 Paper mentioned a tradeoff between average accuracy and worst-group accuracy. Could you provide a more detailed analysis on why this tradeoff occurs and its implications?**
>
> Note that this tradeoff is not our new observation, and was also observed in previous works (e.g. JTT, GroupDRO). A more thorough analysis is stated in GroupDRO's paper, and we would like to borrow its original paragraph for explanation here: "We conclude that regularization—preventing the model from perfectly fitting the training data—does matter for worst-group accuracy. Specifically, it controls the generalization gap for each group, even on the worst-case group. ... Since no regularized model can perfectly fit the training data, ERM and DRO models make different training trade-offs: ERM models sacrifice worst-group for average training accuracy and therefore have poor worst-group test accuracies, while DRO models maintain high worst-group training accuracy and therefore do well at test time." Bam follows the previous insights on the use of regularization and therefore leads to a similar observation in the tradeoff.
>
> **Q11 What are the benefits of introducing controlled datasets like Colored-MNIST, Controlled-Waterbirds, and CIFAR10-MNIST over the standard benchmarks?**
>
> As explained in Section 6.1, standard benchmarks including Waterbirds, CelebA, CivilComments-WILDS, and MultiNLI test the performance of Bam compared with other methods, but we were not able to comprehensively test our algorithm on different class and group imbalance ratios. Therefore, we manually created controlled datasets using Waterbirds, Colored-MNIST, and CIFAR10-MNIST, by varying the dataset sizes, class imbalance ratios, and group imbalance ratios. We visualized the auxiliary variables during the first stage, as shown in Figures 3, 4, 8, and 9. We also used these controlled datasets to test the effectiveness of ClassDiff, as shown in Figures 10-13. Appendices A, B, and E provide dataset details, training details, and analyses.
>
>
> **Q12 Why is Bam's performance, unlike Jtt's, unaffected by full convergence in Stage 1?**
>
> With an appropriate choice of $\lambda$, the learned "bias" is stable when training converges in Stage 1 because the auxiliary variables can help fit the label and make the training loss easier to converge. A larger $\lambda$ within a certain range generally makes the performance of Bam more stable with respect to the training time in Stage 1.
>
> **Q13 How does Bam's computational cost and training time compare with the other mentioned methods?**
>
> When implementing BAM on larger-scale datasets, the auxiliary variables can be stored on the disk instead of in memory. During training, we can load only the auxiliary variables corresponding to the current mini-batch, update these variables, and save their new values on the disk. The additional memory usage is negligible because auxiliary variables are in a much smaller dimension than the raw input. The additional training time is also negligible as we observe that BAM and JTT have almost the same per-iteration running time.
>
> Moreover, we observe that the algorithmic changes from JTT to Bam can lead to faster convergence, resulting in a shorter wall-clock time. For example, Figure 2 shows that the optimal performance of Bam is achieved at T=20 compared with JTT's T=60, resulting in 22min 42s shorter amount of time in Stage 1 using the exact same machine (11min 27s for Bam and 34min 9s for JTT).
>
> **Q14 Effect of Imbalance: How does the class or group imbalance in datasets affect the performance of Bam?**
>
> In our paper, we conduct extensive experiments under various kinds of group and class imbalances, and Bam consistently achieves a good performance, as shown in Section 6.
>
>
> **Q15 Do the datasets used for the experiment (e.g., Waterbirds, CelebA, MultiNLI, and CivilComments-WILDS) truly reflect real-world distributions and biases? What are the potential risks if the model is applied in the real world?**
>
> We use the four benchmark datasets for the main experiments due to their popularity in related literature on robustness. These are carefully crafted datasets that have received high (2,000+) citations. Real-world data can indeed be more complex and have other characteristics and biases, but our proposed method Bam, as well as previous methods like JTT, CNC, etc., should in any case at least do better than plain ERM in terms of worst-group robustness.

---

> ### Author Response · Authors · 2023-11-29
> **Response to Reviewer PXZp (Part IV)**
>
> **Q16 The experiment mentions an improvement in the accuracy of "worst-group". Have the authors considered whether such improvements might exacerbate the marginalization of niche groups in some cases?**
>
> In fact, the objective of improving the "worst-group" is to exactly alleviate the marginalization of underrepresented (minor) groups. The goal here is to ensure every group has good performance, instead of only the dominant ones. It is unlikely that such improvements in "worst-group" will exacerbate the marginalization.
>
> **Q17 Bam has been compared to CNC methods using contrast learning in some experiments. Considering that contrast learning may amplify the importance of certain features, how might this affect the model's performance in terms of diversity or inclusion?**
>
> Note that Bam and CNC are both two-stage training algorithms where Bam aims at improving Stage 1 through bias amplification and CNC aims at improving Stage 2 through constrastive learning. In the paper, we mention the potential of combining CNC's Stage 2 with Bam's Stage 1. This should ideally make the model's performance in promoting diversity and inclusion even better because it pays more attention to underrepresented groups.
>
> **Q18 Are there potential risks for the choice of hyperparameters such as λ and T, such as overfitting or instability due to over-tuning in real-world applications?**
>
> In the paper, we provide detailed analyses of the robustness of these two hyperparameters. As shown in section 6.4, a wide range of λ and T should all yield competitive performance.
>
> **Q19 Bam uses auxiliary variables to aid in training. Are these auxiliary variables likely to lead to some unforeseen bias or misrepresentation in practical applications?**
>
> The auxiliary variables themselves will indeed make the model in Stage 1 more biased (due to its bias amplification effect). Therefore, it is crucial that the model is continued being trained in Stage 2 to correct the bias, where the minority group examples are upweighted using information from Stage 1.
>
> Thank you again for your questions, especially those on broader impact. We will include an additional ethical statement section in the final version of this paper.
>
> References:
>
> - [JTT] Liu et al., 2021. [Just Train Twice: Improving Group Robustness without Training Group Information](https://arxiv.org/pdf/2107.09044.pdf)
>
> - [CNC] Zhang et al., 2022 [Correct-N-Contrast: A Contrastive Approach for Improving Robustness to Spurious Correlations
> ](https://arxiv.org/abs/2203.01517)

---

### Review · Reviewer_eZXj · 2023-11-08

**Summary Of Contributions:**

The paper introduces BAM which is a two-stage training algorithm to improve neural network predictions for minority subgroups when group annotations are limited or missing. In the first stage, BAM uses a bias amplification mechanism to consider biases in the data in model training. In the second stage, called rebalanced training, BAM focuses on misclassified samples from the first stage for further improvement. BAM competes well with existing methods in addressing spurious correlations in computer vision and natural language processing benchmarks. The study also identifies a stopping criterion that reduces the need for group annotations without affecting the accuracy of disadvantaged subgroups. Extensive analyses confirm BAM's effectiveness in addressing bias and improving model accuracy when group annotations are scarce or unavailable.

**Audience:**

Yes

**Claims And Evidence:**

Yes

**Requested Changes:**

1. In Table 3, it seems that the performance is not sensitive with respect to lambda. I would like to see what happens if we set lambda = 0.

2. In Table 4, please include smaller values for lambda, starting from lambda = 0.

3. Since the focus of this paper is on bias, I think it is appropriate that bias metrics such as demographic parity are also reported for comparison.

4. An experiment about the mu parameter can be helpful.

**Strengths And Weaknesses:**

Strengths:

1. The proposed method tackles an important challenge.

2. Experiments are performed on both CV and NLP datasets and demonstrate a consistent advantage in all benchmarks.

3. The implementation is provided and the results seem reproducible.

Weaknesses:

1. The theoretical contribution of the paper is weak and quite trivial.

2. Comparison is mostly based on performance which may not be enough to judge model bias. In the absence of such a comparison, it is not easy to conclude that the proposed method is helpful in reducing bias.

---

> ### Author Response · Authors · 2023-11-29
> **Response to Reviewer eZXj**
>
> Thank you for your review and for appreciating our contributions. We address your questions below.
>
> **Q1 The theoretical contribution of the paper is weak and quite trivial.**
>
> While our paper doesn't come with a rigorous theoretical contribution, we note that spurious correlations in deep learning pose a very challenging theoretical question and thus almost all papers in this area are empirical. We believe we did a comprehensive and solid evaluation of our proposed method.
>
> Furthermore, our investigations reveal interesting properties of the auxiliary variables, which could be of interest to future theoretical studies. In particular, Section 6.2 shows that the auxiliary variables for majority and minority examples are notably different.
>
> **Q2 Comparison on performance may not be enough to judge model bias**
>
> Worst-group accuracy is a well-established criterion and a widely accepted objective in the literature on robustness to spurious correlations. In this context, we believe we did a fair and sufficient comparison to existing methods on the same problem.
>
> Moreover, as mentioned in our response to Q1, we did further investigations on the auxiliary variables going beyond the performance metric, which confirms the intuition behind our method (see Section 6.2).
>
> **Q3 \& Q4 Additional experiments on lambda and sensitivity of mu**
>
> Thank you for raising this question. We have run additional experiments and have updated our manuscript to include the results in Section 6.4 (changes highlighted in red).
>
> We have the following observations: (1) by looking at the updated Table 3 and comparing $\lambda=0$ with the rest of the results, we find that $\lambda>0$ is much better, and thus the auxiliary variable is effective in improving the performance for a wide range of values; (2) by looking at the updated Table 4, we can find the strong effect of the auxiliary variable in preventing the model from overfitting as $\lambda$ gets larger. As $\lambda$ increases (0, 5, 20, 30, 40, 50, 70), the performance at large $T$ keeps increasing. (3) by looking at the updated Table 5, we can find that Bam achieves competitive performance for a wide range of $\mu$'s.
>
> The above set of results further strengthen the efficacy of Bam and we sincerely appreciate your suggestions.
>
> **Q5 Other bias metric such as demographic parity bias**
>
> As mentioned in Q2, worst-group accuracy is widely adopted as the metric in the literature on spurious correlations. Other fairness metrics, including demographic parity as you mentioned, are indeed interesting subjects to study but are focused on different types of biases and therefore are out of the scope of the current paper.

---

> > ### Comment · Reviewer_eZXj · 2023-12-18
> > **Bias metric**
> >
> > What I meant by further comparison is to include bias metrics such as demographic parity, equality of odds, etc. These metrics are particularly developed to measure model bias. "Worst-group accuracy" offers a notion to rely on but I don't think it is as prelevant. Please check Table 1 in this paper: https://dl.acm.org/doi/pdf/10.1145/3457607

---

> > > ### Author Response · Authors · 2023-12-19
> > >
> > > Thank you for the suggestion. We agree that other metrics of fairness and bias are important and it would be interesting to investigate whether the ideas in our paper are helpful for them. We'd like to emphasize that the worst-group accuracy metric is a significant one on its own, and there are a large number of papers in the spurious correlation literature focusing exclusively on worst-group accuracy. Its prevalence is evidenced by the extensive citation of the GroupDRO paper, "Distributionally Robust Neural Networks for Group Shifts: On the Importance of Regularization for Worst-Case Generalization," published at ICLR 2020, with 1,162 citations to date. We will make it more clear in our paper that we focus only on worst-group performance and leave the study on other bias and fairness metrics for future work.

---

### Review · Reviewer_FBUU · 2023-12-03

**Summary Of Contributions:**

1. Target on the spurious correlations, this paper proposes a group-annotation-free approach. This paper uses a two-stage method, firstly amplifying the bias from spurious correlations and then holding rebalanced training.
2. In the Bias Amplification stage, this paper introduces learnable auxiliary variable to realize the Amplification.
3. This paper also proposes a stopping criterion classdiff to select the best model during the rebalanced training

**Audience:**

Yes

**Broader Impact Concerns:**

No ethical issues.

**Claims And Evidence:**

No

**Requested Changes:**

1. The term of “Annotation free“ should be specific as “Group annotation free”
2. The Bias Amplification in stage 1 should be proved by a more intuitionistic approach.
3. The blank results and missing std in Table 1 and Table 2 should be filled.
4. The comparison method [2] and more group-annotation-free approaches (if possible) should also be added.

[1] Sohoni, et al. No subclass left behind: Fine-grained robustness in coarse-grained classification problems. NeuIPS 2020.

[2] Seo, et al. Unsupervised learning of debiased representations with pseudo-attributes. CVPR 2022.

**Strengths And Weaknesses:**

S1. The whole paper is well-written and easy to follow.
S2. The group-annotation-free approach for spurious correlations lacks further work to achieve close performance to the group-annotation-dependent approach and it is of great significance in practice as well.
S3. The idea of bias amplification and the introduction of learnable auxiliary variables are absorbing.
S4. The experiments of the ablation study are well organized. The results are also well presented.

W1. Stage 1 aims to amplify the bias from spurious correlations. However, this paper does not prove that stage 1 indeed amplifies such bias in experiments. More accurate evaluations can be introduced to assess whether the bias is indeed amplified.
W2. Using the simple stopping criterion classdiff cannot be reliable through an intuitive thought. For example, if the bias amplification leads to a heavy drop in the performance of the major group and such performance gets close to the performance of the minor group. The classdiff between the major and minor groups can also be small.
W3. The author mentioned two related works that improve group robustness without any group annotations, i.e. [1] and [2].
In experiments, this paper only compares itself with [1] and does not compare with [2] or other works that do not require group annotations. In addition, this paper does not provide the selection criteria for the comparison approaches.
W4. Samely in the comparison results, some results are provided std while some only contain the mean values or do not provide exact results. This paper does not mention the reasons for the lack of results and std.

[1] Sohoni, et al. No subclass left behind: Fine-grained robustness in coarse-grained classification problems. NeuIPS 2020.

[2] Seo, et al. Unsupervised learning of debiased representations with pseudo-attributes. CVPR 2022.

---

> ### Author Response · Authors · 2023-12-17
> **Response to Reviewer FBUU**
>
> Thank you for your careful reading of our paper and valuable feedback. We address your questions below.
>
> **Q1 “Stage 1 aims to amplify the bias from spurious correlations. However, this paper does not prove that stage 1 indeed amplifies such bias in experiments. More accurate evaluations can be introduced to assess whether the bias is indeed amplified.” & “The Bias Amplification in stage 1 should be proved by a more intuitionistic approach.”**
>
> We set up the whole analysis section (Section 6) to verify and understand the bias amplification effect in Stage 1. Sections 6.1 and 6.2 provide convincing figures showing that the auxiliary variables learn to differentiate between majority and minority samples and that the difference gets clearer as training progresses. We can see that the model relies more heavily on the auxiliary variables when making predictions on the minority examples than the majority examples, and thus the neural network itself is more biased towards the majority samples.
>
> Furthermore, we have looked into the incorrectly classified examples and verfied that our algorithm Bam indeed makes more mistakes on the minority examples than JTT at the end of Stage 1, verifying its bias amplification effect. See table below:
>
>
> | Group Size | Group Description   | # JTT mistakes | # Bam mistakes |
> |------------|---------------------|------|------|
> | 3498       | Waterbird in Water  | 7    | 10   |
> | 184        | Waterbird in Land   | 65   | 80   |
> | 56         | Landbird in Water   | 41   | 42   |
> | 1057       | Landbird in Land    | 116  | 90   |
>
>
> In the table above, we can see that Bam makes more mistakes in the minorty samples (Waterbird in land and Landbird in water) and fewer mistakes in the majority samples (Waterbird in water and Landbird in land) than JTT. A better-quality error set (to be upweighted in the second stage) should better help the model to improve its final robustness, and such intuition is confirmed in the consistently improved performance shown in Tables 1 and 2 in our paper.
>
>
>
> **Q2 The term of “Annotation free“ should be specific as “Group annotation free”**
>
> Thank you for the suggestion. We have changed the terminology per your suggestion.
>
>
> **Q3 Using the simple stopping criterion classdiff cannot be reliable through an intuitive thought.**
>
> We do not claim that ClassDiff is a good proxy for worst-group accuracy in *all* cases, but our extensive experiments show that it works really well in training trajectories that happen in practice. There certainly exist possible corner cases where ClassDiff is small but the worst-group accuracy is bad, but such bad cases do not appear in practice. We performed an extensive ablation study in order to understand the effectiveness and limitations of ClassDiff (Appendix E). We find that ClassDiff always has good anti-correlation with the worst-group accuracy as long as it shows a clear increase/decrease trend during training.
>
> Furthermore, note that very little work has been done in the challenging setting where no group annotations are available, and our Bam algorithm together with the ClassDiff metric achieves the best overall worst-group accuracy. We believe such results are interesting enough to the community and could inspire theoretical studies on ClassDiff in the future.
>
>
> **Q4 The blank results and missing std in Table 1 and Table 2 should be filled.**
>
> Note that the caption of Tables 1 and 2 says "results of other approaches come from their original papers." Such convention was adopted in many previous papers [2-5]. There are some blank results and missing STDs because the cited papers did not report accuracies or STDs on the corresponding datasets.
>
> [1] Sohoni et al., NeurIPS 2020. [No Subclass Left Behind: Fine-Grained Robustness in Coarse-Grained Classification Problems](https://arxiv.org/abs/2011.12945)
>
> [2] Zhang et al., ICML 2022. [Correct-n-Contrast: A Contrastive Approach for Improving
> Robustness to Spurious Correlations](https://arxiv.org/abs/2203.01517)
>
> [3] Nam et al., ICLR 2022. [Spread Spurious Attribute: Improving Worst-group Accuracy with Spurious Attribute Estimation](https://arxiv.org/abs/2204.02070)
>
> [4] Kirichenko et al., ICLR 2023 [Last Layer Re-Training is Sufficient for Robustness to Spurious Correlations](https://arxiv.org/abs/2204.02937)
>
> [5] Liu et al., ICML 2021. [Just Train Twice: Improving Group Robustness without Training Group Information](https://arxiv.org/pdf/2107.09044.pdf)
>
>
> **Q5 The comparison method [6] and more group-annotation-free approaches (if possible) should also be added.**
>
> The group-annotaion-free setting still remains largely underexplored and there are only a handful of papers that consider this setting. We have added the results of [6] to our Tables 1 and 2, where we can see that Bam still achieves the best overall results.
>
> [6] Seo, et al. Unsupervised learning of debiased representations with pseudo-attributes. CVPR 2022.

---

### Decision · Action_Editor_Eho1 · 2024-02-25

**Recommendation:** Accept as is

**Comment:**

The paper underwent a through reviewing process. The revisions made by the authors generally addressed the reviewers' concerns, and finally, all the three reviewers recommended to (weakly) accept this submission. The AE concurs with the reviewers' recommendations.

**Audience:**

The TMLR's audience with interests in worst-group performance, e.g. long-tailed subgroups, will be interested in the findings of this paper.

**Claims And Evidence:**

Yes, the claims made in this submission are fairly supported.